# Analysis of Online Consultations and Emergent Treatments of Operative Dentistry and Endodontics during the COVID-19 Epidemic

**DOI:** 10.3390/ijerph19041931

**Published:** 2022-02-09

**Authors:** Fushi Wang, Weiwei Qiao, Fei Wang, Liuyan Meng

**Affiliations:** 1The State Key Laboratory Breeding Base of Basic Science of Stomatology (Hubei-MOST) and Key Laboratory of Oral Biomedicine Ministry of Education, School and Hospital of Stomatology, Wuhan University, Wuhan 430079, China; fushiwang93@whu.edu.cn (F.W.); weiwei_qiao99@163.com (W.Q.); wf5511@whu.edu.cn (F.W.); 2Department of Cariology and Endodontics, Hospital of Stomatology, Wuhan University, Wuhan 430079, China

**Keywords:** operative dentistry, endodontics, COVID-19 epidemic, online consultations, emergent dental treatments

## Abstract

Background: The purpose of the present study is to evaluate the characteristics of online consultations and emergent dental treatments and analyze the status of diseases related to operative dentistry and endodontics (ODE) during the COVID-19 epidemic. Methods: Online consultations were collected from 3 February to 21 April 2020. The electronic medical record system was accessed to collect clinical diagnoses and emergent dental treatments from 9 January to 21 April 2020. Results: A total of 2419 patients visited us and received treatments 2 weeks before the lockdown. The number of patients decreased to 537 during the 76 days of the lockdown. Among them, dental examinations accounted for the majority of visits (88.83%). After 7 April, the outpatient number increased to 36.79 ± 6.63 per day, but the proportion of dental examinations and treatments did not change significantly. A total of 1218 online consultations were completed before the lockdown. The most common dental problem was pulpitis (48.1%). After 7 April, consultations surged from 23.15 ± 8.54 to 44.43 ± 12.63 per day. Consultations related to pulpitis, apical periodontitis, or dental caries remained stable. Conclusions: Correct understanding, active treatments, and appropriate psychological interventions for the ODE staff during the COVID-19 epidemic are necessary. Our results may provide references to arrange staff and treat patients more efficiently for future epidemics.

## 1. Introduction

Coronavirus disease 2019 (COVID-19), a public health emergency of global concern, is an infectious disease caused by severe acute respiratory syndrome coronavirus 2 (SARS-CoV-2). Like SARS-CoV, SARS-CoV-2 uses the angiotensin-converting enzyme 2 receptor and mainly spreads through the respiratory tract [1]. More than 220 countries and regions have reported the presence of COVID-19, with nearly 300 million confirmed cases and more than 5 million deaths. The growth trend is slowing down due to technical issues with the electronic laboratory reporting system [2]. Although the development of vaccine research brings hope towards the COVID-19 pandemic, the situation of shortage and unfair distribution of vaccines is still not optimistic [3].

The municipal government of Wuhan announced an unprecedented lockdown on 23 January 2020 to protect more people from being infected and avoid the collapse of the medical system. The measures taken during the lockdown have been proved effective since the lockdown delayed the spread of SARS-CoV-2 to other cities by an average of 2.91 days and reduced the number of confirmed cases by hundreds of thousands [4]. Encouraged by the positive role played by Wuhan, several governments responded to the disease with more stringent epidemic prevention measures such as lockdowns of international borders and public spaces or even further extension of the lockdown [5].

Although measures during the lockdown have been proved successful, the lockdown confined the people’s medical needs related to other diseases to some extent. Generally, patients are less tolerant to the diseases associated with operative dentistry and endodontics (ODE) [6], which are some of the major diseases causing pain among various dental diseases [7]. After experiencing pain, patients typically seek medical treatment as soon as possible. Unfortunately, the actual status of ODE-related diseases among patients during the COVID-19 epidemic is still unknown. As the biggest stomatological hospital in Wuhan that mainly provides dental services to patients from the whole Hubei province and other adjacent provinces, our hospital established an online consultation platform and emergent dental treatments to meet the dental needs during the epidemic.

The present study aimed to evaluate the characteristics of online consultations and emergent dental treatments and analyze ODE-related diseases’ status during the lockdown due to the COVID-19 epidemic. We would like to share our analysis and experiences from our hospital, hoping that the relevant findings provide references to treat the patients more efficiently if the ODE staff encounters a similar situation in the future.

## 2. Materials and Methods

### 2.1. Ethical Considerations

The present study was approved by the Ethics Committee of the Hospital of Stomatology, Wuhan University (No.2020-B33). The study complied with the ethical principles laid out in the Declaration of Helsinki: informed consent, privacy, and confidentiality.

### 2.2. Data Collection

The department of Cariology and Endodontics was selected due to its deep engagement in emergency dental treatment during the epidemic. Its outpatient number and number of online consultations for ODE-related diseases were the highest in our hospital. Its opening hours are from 8:00 a.m. to 5:00 p.m. each day.

The user terminal of the online consultation platform was based on the WeChat mini program (Tencent Inc., Shenzhen, China), which is one of the most commonly used instant messaging applications among Chinese mainlanders and the overseas Chinese population. Consultations comprised dialogues, photographs, voices, videos, and even emoticons. After each consultation, patients were asked to score from 1 to 5 based on their satisfaction with the whole consultation process (a score of 5 denoted the most satisfying experience).

The back-end terminal was based on a program developed by Leantech (Jingchen Technology Co., Ltd.; Shanghai, China). It was managed by the network information center of our hospital. Dentists installed the program on their personal computers with built-in or external cameras in advance. In addition, consultations could be easily browsed and looked up even after finishing.

### 2.3. Participants

Body temperature detection was compulsory for emergent dental treatments, and a COVID-19 epidemiological questionnaire was distributed before entree to the hospital. The questionnaire was reported as Meng et al. [8]. It consisted of the following four questions:Are you a confirmed or suspected COVID-19 patient?Are you a confirmed or suspected COVID-19 patient without symptoms?Have you recently been showing symptoms related to COVID-19, such as fever, dry cough, fatigue, and dyspnea?Have you been in contact with confirmed or suspected COVID-19 patients recently?

Only the patients who answered the questionnaire and underwent body temperature measurements were allowed to enter the hospital. At the reception of our department, the same procedures were repeated for confirmation. Patients with non-emergency dental symptoms were suggested postponement of the treatment by the dentists. Therefore, most of the non-emergency medical records were excluded. There was also a situation where non-emergency patients insisted on treatment. In this case, only patients with a negative nucleic acid certificate within 7 days could receive treatment.

The consultation platform was launched on 3 February 2020, less than two weeks after the lockdown. It was available from 8:00 a.m. to 5:00 p.m. every day. Our department’s medical records of emergent dental treatments were collected for better comparison accuracy. Online dialogues of 79 days and medical records of 104 days (until 21 April 2020; 2 weeks after the lockdown) were reviewed. Diagnoses and treatments were recorded and classified.

Additionally, to simplify the data from an extensive period, we merged the online consultation data every week for comparison, except the data from the last two days of the lockdown.

### 2.4. Data Analysis

The analysis of the qualitative findings was performed with the software GraphPad Prism 9.0 (USA), which allowed for the organization and analysis of the results and increased accuracy.

## 3. Results

As shown in Figure 1, female patients were the majority, regardless of the phase of online consultations or dental treatments. For offline treatment, patients of 30–40 years old were the majority at any phase. Concerning the distribution of age for online consultation, the age groups of 30–40 years old (30.38%), 40–50 years old (23.65%), and 20–30 years old (21.92%) accounted for the majority.

According to the back-end data server record, our department performed 1218 online consultations for ODE-related diseases from 3 February to 7 April 2020. All consultations were successfully classified every week with no overlap. The number of online consultations was more than 100 (130.67 ± 14.01 per week) during the 9 weeks. Although the data from the last 2 days were reported separately, they were consistent with the data from the previous 9 weeks when converted into weekly data. Pulpitis was the most common disease, which accounted for approximately half (48.7%) of all primary diagnoses (Figure 2). Apical periodontitis and dental caries accounted for 21.05% and 18.16%, respectively. These three diseases were the most frequent among ODE-related online diagnoses. In contrast, the incidence of non-carious cervical lesions, dentinal hypersensitivity, and cracked teeth was relatively low among total online consultations. The category “Others” included diseases not classified due to a lack of clinical or radiological examinations, rare diseases such as traumatic injuries, diseases referred to other departments, or invalid replies.

After the lockdown, consultations surged from 23.15 ± 8.54 per day to 44.43 ± 12.63 per day. The number of consultations for pulpitis, apical periodontitis, or dental caries remained stable. Meanwhile, inquiries about the official resumption of our hospital constituted the majority.

Altogether, 537 patients visited our department during the lockdown for emergent dental treatments. For the 78-day lockdown period, there were 33 days that no patient visited our department.

Figure 3 shows the percentages of diagnoses among ODE-related dental emergency treatments during three different periods. The rate of dental caries dramatically decreased (from 19.76% to 2.98%) and subsequently tripled (from 2.98% to 9.32%) from Phase I to Phase III. In contrast, the percentages of pulpitis and apical periodontitis cases did not change significantly. Among the category “Others”, the rates of tooth fracture and cracked tooth cases increased slightly. The percentages of other ODE-related diagnoses showed variable changes. However, when they were converted to the actual number, all diagnoses decreased considerably due to the sharp drop in outpatients during the lockdown.

Figure 4 compares the percentages of various ODE-related dental emergency treatments performed during the three phases. The rates of root canal treatments and resin restorations decreased significantly during the initial 2 phases (from 39.43% to 8.94% and from 21.85% to 1.12%, respectively). In contrast, the percentage of dental examinations increased rapidly (from 31.97% to 88.83%) and subsequently remained high (89.51%). Moreover, other ODE-related procedures decreased to approximately 0%, and the lockdown’s end did not significantly influence this proportion.

During three different periods, the daily number of outpatients decreased from 172.79 ± 72.83 to 7.07 ± 8.91 and subsequently increased to 36.79 ± 6.63. Additionally, the number of outpatients in the first two weeks after the lockdown did not return to the average level.

## 4. Discussion

Two weeks before the lockdown, dental patients were treated in the usual manner at our hospital, although the news about SARS-CoV-2 had been reported since December 2019. Diagnoses and treatments during these days consisted of routine dental procedures and were consistent with the average number of outpatients in our department. However, citizens became vigilant on 20 January 2020, when the human-to-human transmission of COVID-19 was confirmed. Although quarantine measures were not implemented immediately, the daily number of outpatients in the department on 20 January, 21 January, and 22 January were 190, 99, and 48, respectively, showing a 50% decrease every day. The ensuing lockdown influenced citizens’ ordinary lives and medical needs.

It was recommended that all routine dental procedures be postponed during the epidemic except dental emergency cases [9]. Our hospital established an isolation clinic and gradually increased the protection level to protect the staff and serve the emergency patients [8]. After strict pre-check triaging, oral examination procedures were performed, and aerosol generation was minimized using rubber dams, negative pressure saliva ejectors, and the four-handed technique. Strict surface sterilization and environmental sterilization were also prerequisites.

More women received dental treatment or consulted on the platform, which embodied that these women possessed stronger health consciousness and more vital ability of execution [10]. In addition, patients below 70 years accounted for the absolute majority, especially in online consultations. This demonstrated that the current digitalized epoch was not friendly to the elderly, which reduced their medical need [11].

The percentage of cases with dental caries among diseases requiring emergency dental treatments decreased sharply due to the lockdown, indicating that dental caries with mild symptoms may not force the patients to visit the hospital during the outbreak. In contrast, the percentages of pulpitis and apical periodontitis cases did not change significantly. However, the rates of tooth fracture and cracked tooth cases increased with the epidemic’s progression. Direct impact from external forces is a common cause of traumatic injuries. Citizens may preferentially focus on food during the quarantine. Chewing hard food more frequently and paying less attention to dental care could lead to a higher incidence of tooth fracture or cracked teeth [12].

Although patients left no stone unturned to get to the hospital, there was a high probability that they would not be treated immediately, as observed by the high percentage of dental examinations. The unknown aspects of COVID-19, the lack of high-quality personal protective equipment, and the continuously increasing number of confirmed cases or deaths exerted enormous pressure on the dental staff, leading to anxiety and panic. It was reported that the medical staff was more prone to develop psychosocial problems during the epidemic [13]. Meanwhile, the strike rate and the demission rate of medical staff grew during the outbreak [14]. In addition, the dental practice exposed the dental staff to tremendous risk of COVID-19 due to face-to-face conversation, exposure to saliva, blood, or other body fluids, and the risk of pricks from sharp dental instruments [15]. These reasons forced the dentists into opting for more passive treatments.

Regarding the detection methods of SARS-CoV-2, polymerase chain reaction (PCR) testing and antibody detection have advantages and disadvantages [16]. PCR test results earlier in the disease course were more likely to be positive, even when cutaneous manifestations rather than systemic symptoms defined the date of onset. On the contrary, rapid tests for antibodies have been widely developed commercially but are of varying quality. Many manufacturers do not disclose the nature of the antigen used. These tests are purely qualitative and can only indicate the presence or absence of SARS-CoV-2 antibodies. In China, PCR testing is officially preferred. By 27 June 2020, the official data released by the Chinese medical and health institutions have cumulatively carried out more than 90 million SARS-CoV-2 PCR tests [17]. For these reasons, our hospital required non-emergency patients who insisted on treatment to show a negative nucleic acid test report within 7 days. This was responsible for the patient and made medical workers perform dental diagnoses and treatment with reassurance.

Accordingly, dentists had to adopt conservative measures whenever possible, which included prescribing anti-inflammatory analgesics and instructing about the correct method of oral hygiene rather than performing radical procedures. It was also reported that acetaminophen might be better than ibuprofen, as ibuprofen may influence immune function [18]. Although pharmacological management in antibiotics or analgesics was considered an alternative [19], drug resistance due to long-term consumption of antibiotics during the COVID-19 epidemic should receive serious attention. Antimicrobial resistance is an inevitable evolutionary response after the usage of antibiotics. Thus, the risks of antibiotic resistance are increasing, and multi-drug resistance in organisms could severely restrain therapeutic options [20].

Besides dental treatments, reduplicative mood-pacification became more critical, as patients were not confirmed or suspected. COVID-19 patients were also under insurmountable psychological pressure, leading to various psychological problems, such as anxiety, fear, depression, and insomnia [21]. Most of the patients would comprehend and follow the dentists’ advice, but it was possible that some of them did not coordinate with the dentists or were even involved in disputes with the dentists. Patient communication would be essential to prevent such incidences [22].

Since citizens were instructed to quarantine themselves and avoid unnecessary contact, the “at your fingertips” medical services have become convenient. They could obtain professional medical consultations from their homes. In addition to toothaches, patients consulted online dentists with questions, not requiring them to visit the hospital. Such queries included brushing their teeth properly or which toothpaste was better. Dentists could also answer many questions simultaneously, which was more efficient than usual. Some patients consulted with revisit-related queries as the outbreak disrupted their predetermined revisit schedule. After the lockdown, online consultations could help the patients with oral health education who do not need to go to the hospital. This approach could also instruct the patients who do not know which department to visit. In this case, there will be much time and money being saved.

As shown in Figure 2 and Figure 3, the percentages of pulpitis and apical periodontitis cases diagnosed clinically before the lockdown, and the proportion of online consultations for the same diseases was consistent. It was speculated that the distribution of ODE-related diseases among online consultations could reflect the actual status of ODE-related diseases among the population during the epidemic to some extent. However, it was essential to clarify that the diagnosis obtained through online consultation was just a primary diagnosis. Due to the method’s limitations, online consultations still had many disadvantages. Although photographs, dialogues, videos, and voices were optional, clinical examinations and radiographs were indispensable for diagnosing specific ODE-related diseases. Though online consultations are not perfect, they are well suited for conditions in which infrastructure remains intact and clinicians can see patients [23]. More diseases could be differentiated while comparing the data of online consultations and emergent dental treatments. Due to the lack of adequate intervention, dentists could only analyze the diseases roughly, advise patients to visit the oral emergencies whenever appropriate, and instruct them about oral hygiene and the rational use of drugs. More importantly, for patients suffering from severe pain due to deep caries, pulpitis, apical periodontitis, or crown fractures, emergency treatment following standard, contact, and airborne precautions would be necessary if medicine failed to relieve the pain [19,24].

The Wuhan city was unblocked on April 8. At the same time, the medical order was gradually restored, and the outpatient number was gradually increased. Consequently, the ODC platform was unavailable and did not provide services. The patient’s medical needs were met, but their counseling needs were not. A more common and intelligent ODC platform is necessary whenever providing more dental counseling services or with the outbreak of SARS-CoV-2 Variant Delta (B. 1.617. 2) or Omicron (B. 1.1. 529) [25].

## 5. Conclusions

We analyzed the data and characteristics of online consultations and emergent dental treatments at our department during the COVID-19 epidemic. Online consultations could attend to simple inquiries. They could also enable triage to a certain extent, making full use of limited medical resources. On the other hand, emergent dental treatments could relieve patients’ pain when dentists were provided with high-quality personal protective equipment. Correct understanding, active treatment measures, and appropriate psychological interventions during the COVID-19 epidemic are necessary for the ODE staff with the gradual resumption of work. The data analysis from our department could provide references to arrange staff duties and treat patients efficiently if a similar epidemic is encountered in the future.

## Figures and Tables

**Figure 1 ijerph-19-01931-f001:**
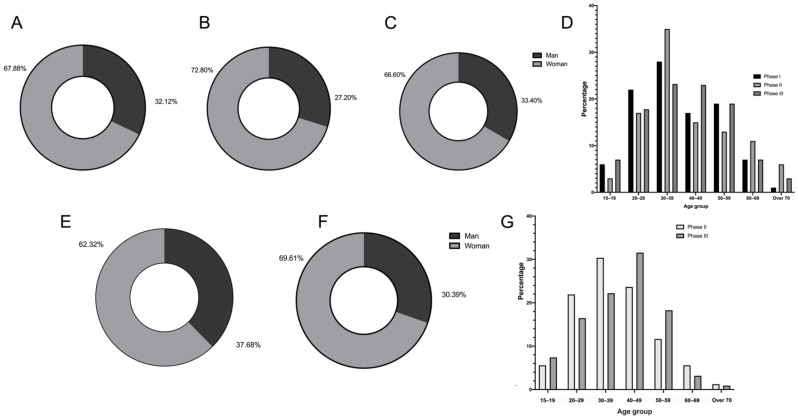
Distribution of genders and ages among online consultants. (**A**) Gender ratio of Phase I of dental treatment. (**B**) Gender ratio of Phase II of dental treatment. (**C**) Gender ratio of Phase III of dental treatment. (**D**) Age distribution among 3 phases of dental treatment. (**E**) Gender ratio of Phase II of online consultation. (**F**) Gender ratio of Phase III of d online consultation. (**G**) Age distribution among 2 phases of online consultation. Phase I: 9 January to 22 January 2020; Phase II: 23 January to 7 April 2020; Phase III: 8 April to 21 April 2020.

**Figure 2 ijerph-19-01931-f002:**
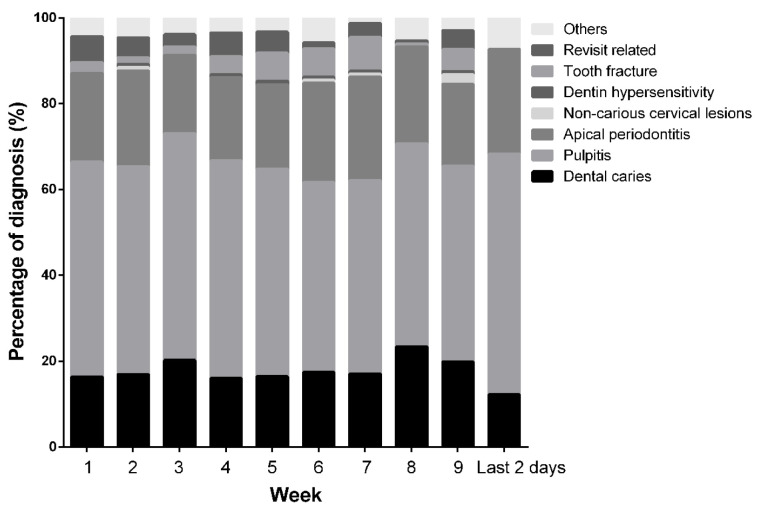
Stacked bar chart of the percentages of online diagnoses during the lockdown.

**Figure 3 ijerph-19-01931-f003:**
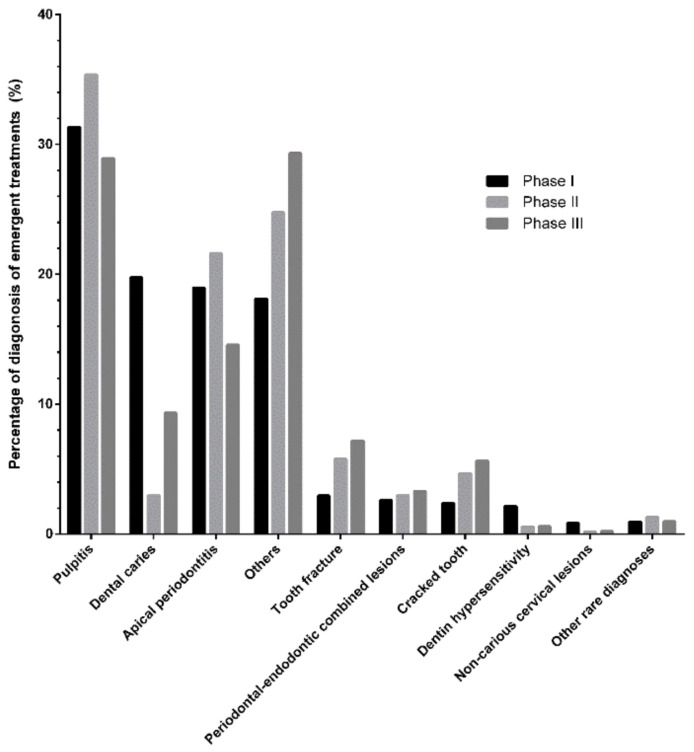
Percentages of ODE-related diagnoses during three periods (rare ODE-related diagnoses included external root resorption, pulp calcification, and other rare diagnoses). Phase I: 9 January to 22 January 2020; Phase II: 23 January to 7 April 2020; Phase III: 8 April to 21 April 2020. ODE: operative dentistry and endodontics.

**Figure 4 ijerph-19-01931-f004:**
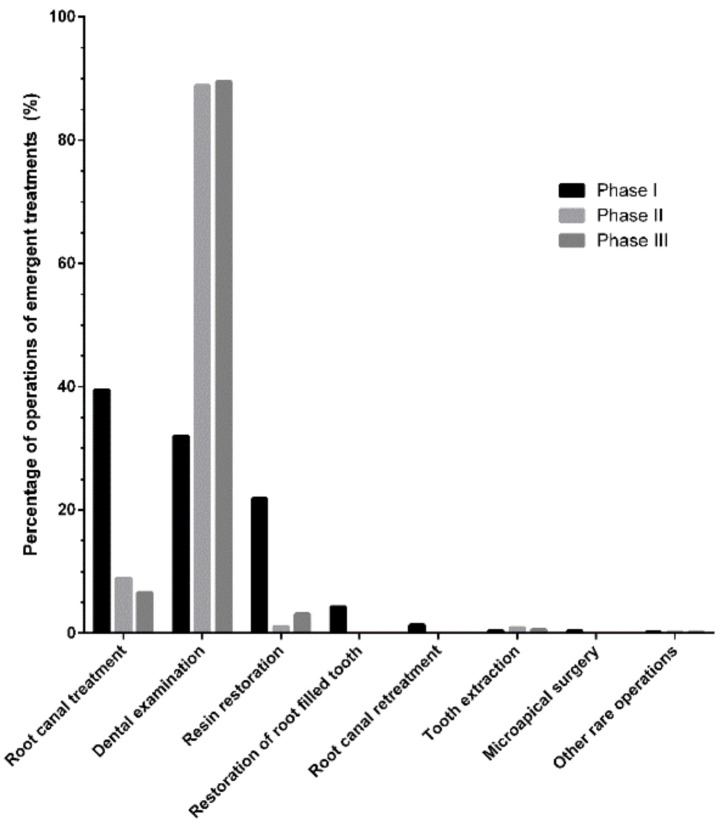
Percentages of emergent dental procedures for ODE-related diseases during three periods (rare procedures included occlusal adjustment, apical barrier technique, and other rare procedures). Phase I: 9 January to 22 January 2020; Phase II: 23 January to 7 April 2020; Phase III: 8 April to 21 April 2020. ODE: operative dentistry and endodontics.

## Data Availability

The raw/processed data required to reproduce these findings cannot be shared at this time as the data also forms part of an ongoing study.

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
