# Peer review of "Analysis of Online Consultations and Emergent Treatments of Operative Dentistry and Endodontics during the COVID-19 Epidemic"

_ijerph, 2022, doi:10.3390/ijerph19041931_

Round 1

Reviewer 1 Report

Materials and methods. 

  1. the ethical approval should be separated as a subpoint, with the specification of which hospital, which university board? please follow MDPI guidelines.
  2. what was the body temperature limit?
  3. I think that COVID-19 test; antibody or PCR should be performed but I did not read about it in the text. How is it possible that there is no data about it? If there is an explanation, please add it. 
  4. Was it possible to have patients from other clinics as well as referred patients?
  5. What is the statistics of the present patient about body temperature, gender ratio, age cohort? These parameters are needed. I am sure you have statistics about it, please include them.
  6. I think it is also very interesting to add how many colleagues were on shift for these procedures, and how long was the clinic opened. Only day or night shifts were available. 
  7. Please indicate that if there were pulpitis, what was the procedure? Trepanation and provisional, was there any follow-up, or recall?
  8. How was the online consultation performed, who analysed or answered the questionnaires? Dentists, dental students, dental nurses? and why? Were there any previous examples?

Author Response

Manuscript ID: ijerph-1549670
Title: Analysis of online consultations and emergent treatments of operative dentistry and endodontics during the COVID-19 epidemic

The authors would like to thank the editor and the reviewer for their critical and helpful comments and suggestions. These comments are all valuable and beneficial for revision and improvement. All comments have been studied carefully, and the revised manuscript reflects a significant revision that addresses the comments, suggestions, and recommendations for improvement provided by the editor and reviewers. Changes in the revised manuscript are highlighted in red color. We hope the updated manuscript could meet the requirement of approval. The main corrections and responses to the comments are given in a point-by-point manner below:

Reviewer 1:

  1. the ethical approval should be separated as a subpoint, with the specification of which hospital, which university board? please follow MDPI guidelines.

Response: Thank you for your kind advice. We read the ‘Research and Publication Ethics’ in MDPI guidelines carefully. The ethical approval has been separated as a subpoint.

Revised words: 2.1. Ethical Considerations

The present study was approved by the Ethics Committee of the Hospital of Stomatology, Wuhan University (No.2020-B33). The study complied with the ethical principles laid out in the Declaration of Helsinki: informed consent, privacy, and confidentiality. (Section 2.1., Page 2)

  1. what was the body temperature limit?

Response: Thank you for your question. It has been reported that a body temperature of more than  37.5  °C would be a risk factor for aggravation of COVID-19 symptoms from asymptomatic-mild to severe. (Chang, M. C.; Park, Y.-K.; Kim, B.-O.; Park, D., Risk factors for disease progression in COVID-19 patients. BMC infectious diseases, 2020, 20, (1), 1-6.) If we found a patient whose body temperature was more than  37.5  °C, we would ask them to go to the fever clinic.

  1. I think that COVID-19 test; antibody or PCR should be performed but I did not read about it in the text. How is it possible that there is no data about it? If there is an explanation, please add it. 

Response: We appreciate your valuable questions. Hospital of Stomatology, Wuhan University is a specialized stomatological hospital. It cannot test SARS-CoV-2. In addition, patients with negative nucleic acid certificates within 7 days would be recommended. Relevant contents have also been added in the corresponding parts of the manuscript.

Revised words: There was also a situation where non-emergency patients insisted on treatment. In this case, only patients with a negative nucleic acid certificate within 7 days could receive treatment. (Line 5, Paragraph 2, Section 2.3., Page 2)

Regarding the detection methods of SARS-CoV-2, polymerase chain reaction (PCR) testing and antibody detection have advantages and disadvantages [16]. PCR test results earlier in the disease course were more likely to be positive, even when cutaneous manifestations rather than systemic symptoms defined the date of onset. On the contrary, rapid tests for antibodies have been widely developed commercially but of varying quality. Many manufacturers do not disclose the nature of the antigen used. These tests are purely qualitative and can only indicate the presence or absence of SARS-CoV-2 antibodies. In China, PCR testing is officially preferred. By June 27, 2020, the official data released by the Chinese medical and health institutions have cumulatively carried out more than 90 million SARS-CoV-2 PCR testing [17]. For these reasons, our hospital re-quired non-emergency patients who insisted on treatment to show a negative nucleic acid test report within 7 days. This was responsible for the patient and made medical workers carry out dental diagnosis and treatment with reassurance. (Paragraph 5, Discussion, Page 7)

  1. Was it possible to have patients from other clinics as well as referred patients?

Response: Thank you for your questions. In the field of dentistry, we advocate the primary responsibility system. If a dentist initially treated a patient, then the subsequent treatments should be completed by the same dentist. However, as one of the top stomatological hospitals in Wuhan that mainly provides dental services to patients from the whole Hubei province and other adjacent provinces, there are a number of incurable patients from other clinics or other hospitals completing follow-up treatments in our hospital.

  1. What is the statistics of the present patient about body temperature, gender ratio, age cohort? These parameters are needed. I am sure you have statistics about it, please include them.

Response: Thank you for your advice. Regarding the body temperature, every patient included in this study has passed the triage of our hospital. Therefore, the number of patients whose body temperature was beyond the limit was not accounted for. In addition, the diagram of gender ratio and age cohort was added.

Revised figure and words:

Figure 1. Distribution of genders and ages among online consultants. A: Gender ratio of Phase I of dental treatment. B: Gender ratio of Phase II of dental treatment. C: Gender ratio of Phase III of dental treatment. D: Age distribution among 3 phases of dental treatment. E: Gender ratio of Phase II of online consultation. F: Gender ratio of Phase III of d online consultation. G: Age distribution among 2 phases of online consultation.

Phase I: January 9 to January 22, 2020; Phase II: January 23 to April 7, 2020; Phase III: April 8 to April 21, 2020 (Page 4)

As shown in Figure 1, the female was the majority, no matter at any phase of online consultations or dental treatments. For offline treatment, patients of 30-40 years old were the majority at any phase. Concerning the distribution of age for online consultation, the age groups of 30-40 years old (30.38 %), 40-50 years old (23.65 %), and 20-30 years old (21.92 %) accounted for the majority. (Paragraph 1, Results, Page 3)

More women received dental treatment or consulted in the platform, which embodied that these women possessed stronger health consciousness and more vital ability of execution [10]. In addition, patients below 70 years accounted for the absolute majority, especially in the online consultation. It demonstrated that the current digitalized epoch was not friendly to the elderly, which would reduce their medical need [11]. (Paragraph 3, Discussion, Page 6)

  1. I think it is also very interesting to add how many colleagues were on shift for these procedures, and how long was the clinic opened. Only day or night shifts were available. 

Response: Thank you for your good question. Our hospital opens from 8:00 am to 5:00 pm from October to May. Although there is more than 1000 medical staff in our hospital, we only arrange one dentist and one nurse for the night shift due to the particularity of the stomatological industry, except the ward. The platform was available from 8:00 am to 5:00 pm, either, regarding the online consultation. Relevant words have been added to the manuscript.

Revised words: Its opening hours are from 8:00 am to 5:00 pm each day. (Line 4, Paragraph 1, Section 2.2., Page 2)

It was available from 8:00 am to 5:00 pm every day. (Line 2, Paragraph 3, Section 2.3., Page 3)

  1. Please indicate that if there were pulpitis, what was the procedure? Trepanation and provisional, was there any follow-up, or recall?

Response: Thank you for your question. If a patient suffered from pulpitis, they would receive root canal therapy. The whole treatment cycle would sustain for 3 to 4 weeks. And the patient has to revisit the dentist every week.

  1. How was the online consultation performed, who analysed or answered the questionnaires? Dentists, dental students, dental nurses? and why? Were there any previous examples?

Response: We appreciate your valuable question. It was the certificated dentist that performed the online consultation. In general, Online consultations require a strong, reliable broadband connection and mobile communication technology of the fourth generation or the fifth generation (4G or 5G) or long-term evolution standard to meet the essential requirements of voice and video stability and bandwidth restrictions. Therefore, there are several online medical platforms available within 10 years, such as Ask Top Doctor's Advice 24*7, or Chunyuyisheng. However, according to the information we collected, our online medical consultation platform was the first online platform service based on the characteristics of the epidemic.

Reviewer 2 Report

The main interesting point raised is the online triage undertaken by the authors during the pandemic.  It would be valuable to determine how much time and money was saved by using this approach - rather than document numbers of patients attending.  It is obvious that the number of patients attending during a pandemic would reduce but the reader would be more interested in the additional benefits from your management of the pandemic.  Also please quantify how many of the staff were affected negatively by the pandemic - 

Author Response

Manuscript ID: ijerph-1549670
Title: Analysis of online consultations and emergent treatments of operative dentistry and endodontics during the COVID-19 epidemic

The authors would like to thank the editor and the reviewer for their critical and helpful comments and suggestions. These comments are all valuable and beneficial for revision and improvement. All comments have been studied carefully, and the revised manuscript reflects a significant revision that addresses the comments, suggestions, and recommendations for improvement provided by the editor and reviewers. Changes in the revised manuscript are highlighted in red color. We hope the updated manuscript could meet the requirement of approval. The main corrections and responses to the comments are given in a point-by-point manner below:

The main interesting point raised is the online triage undertaken by the authors during the pandemic.  It would be valuable to determine how much time and money was saved by using this approach - rather than document numbers of patients attending.  It is obvious that the number of patients attending during a pandemic would reduce but the reader would be more interested in the additional benefits from your management of the pandemic.  Also please quantify how many of the staff were affected negatively by the pandemic - 

Response: Thank you for your critical suggestions. During the lockdown period, online consultation was the only choice for most of the patients. Therefore, the amount of time and money saved via this approach was not important. However, after the lockdown, online consultations could help the patients with oral health education who do not need to go to the hospital. This approach could also instruct the patients who do not know which department to visit. In this case, there will be much time and money being saved. In addition, we have quantified the exact number of staff who were affected positively by the pandemic. As Meng et al. reported, there were 8 staff and 1 student confirmed with COVID-19 in our hospital. (Meng, L.; Hua, F.; Bian, Z., Coronavirus disease 2019 (COVID-19): emerging and future challenges for dental and oral medicine. Journal of Dental Research 2020, 99, (5), 481-487.) The words were also modified in the manuscript.

Revised words: After the lockdown, online consultations could help the patients with oral health education who do not need to go to the hospital. This approach could also instruct the patients who do not know which department to visit. In this case, there will be much time and money being saved. (Line 8, Paragraph 5, Discussion, Page 7)
